

# Effect of GnRH agonist down-regulation combined with hormone replacement treatment on reproductive outcomes of frozen blastocyst transfer cycles in women of different ages

Jianghuan Xie, Jieqiang Lu and Huina Zhang

Department of Obstetrics and Gynecology, The Second Affiliated Hospital and Yuying Children's Hospital of Wenzhou Medical University, Wenzhou, China

## ABSTRACT

**Objective**. To investigate the effect of GnRH agonist (GnRH-a) down-regulation prior to hormone replacement treatment (HRT) to prepare the endometrium in frozen embryo transfer (FET) cycles in women of different ages.

**Methods**. This was a retrospective study, and after excluding patients with adenomyosis, endometriosis, severe endometrial adhesions, polycystic ovary syndrome (PCOS), and repeated embryo implantation failures, a total of 4,091 HRT cycles were collected. Patients were divided into group A (<35 years old) and group B (≥35 years old), and each group was further divided into HRT and GnRHa-HRT groups. The clinical outcomes were compared between groups.

**Results**. There was no statistically significant difference in clinical outcomes between the HRT and GnRHa-HRT groups among women aged < 35 years. In women of advanced age, higher rates of clinical pregnancy and live birth were seen in the GnRHa-HRT group. Logistic regression analysis showed that female age and number of embryos transferred influenced the live birth rate in FET cycles, and in women aged ≥ 35 years, the use of GnRH-a down-regulation prior to HRT improved pregnancy outcomes.

**Conclusions**. In elderly woman without adenomyosis, endometriosis, PCOS, severe uterine adhesions, and RIF, hormone replacement treatment with GnRH agonist for pituitary suppression can improve the live birth rate of FET cycles.

# INTRODUCTION

Frozen embryo transfer (FET) is a widely used assisted reproduction technique that allows for the storage and utilization of excess embryos, has high clinical pregnancy and live birth rates, and reduces the risk of ovarian hyperstimulation syndrome (OHSS) (*Mackens et al., 2017*; *Veleva et al., 2009*). The optimal endometrial preparation method in FET cycles is controversial. The most commonly used protocol is hormone replacement treatment (HRT), providing flexibility in the timing of medication and embryo transfer.

Corresponding author
Huina Zhang, huina.zh201@163.com

In this protocol, the endometrium is artificially prepared through the administration of exogenous estrogen (E2) and progesterone.

Gonadotropin-releasing hormone agonist (GnRH-a) is a gonadotropin-releasing hormone analog, it can cause an initial flare-up effect on pituitary follicle stimulating hormone (FSH) and luteinizing hormone (LH) secretion. GnRH receptors can be downregulated by GnRH-a leading to the inhibition of gonadotropin secretion and avoiding follicular development to reduce the adverse effects on embryo implantation (*El-Toukhy et al., 2004*); Also, GnRH-a improves pregnancy outcomes by increasing endometrial receptivity (*Orvieto et al., 2008*). However, it remains unclear whether GnRH-a pretreatment is necessary before HRT. A randomized clinical trial indicated that HRT with GnRH-a pretreatment reduced cycle cancellation and improved clinical pregnancy and live birth rates (*El-Toukhy et al., 2004*). However, one study suggested that removing pretreatment of GnRH-a didn't affect pregnancy outcomes and reduced cost and adverse effects related to GnRH agonists (*Samsami, Chitsazi & Namazi, 2018*). It is worth noting that the inclusion criteria, drug doses, and routes of administration varied among studies.

Some studies indicated that HRT + GnRH-a had benefits for specific populations. HRT-FET with GnRH-a pretreatment showed significant improvements in pregnancy outcomes in women with endometriosis or adenomyosis (*Niu et al., 2013*; *Park et al., 2016*). This can be achieved by suppressing the proliferation of pathological lesions and reducing the effects of cytotoxic cytokines (*Khan et al., 2010*; *Tamura et al., 2014*). Additionally, *Wang et al. (2022)* found that pretreatment with GnRH-a can improve live birth rates and reduce the risk of preterm birth for women with polycystic ovary syndrome (PCOS). One study found pretreatment with GnRH-a might improve the ongoing pregnancy rate in hyperandrogenic PCOS women (*Tsai et al., 2017*). Also, HRT + GnRH-a improved the live birth rate and other clinical outcomes, especially for women with PCOS (*Xie et al., 2018*). It is stated that the long-term deleterious effect of estrogen alone, hyperandrogenic environment, hyperinsulinemia and high LH levels can lead to abnormal endometrial receptivity (*Jiang & Li, 2022*; *Lee et al., 2019*; *Schulte, Tsai & Moley, 2015*; *Wiwatpanit et al., 2020*). The dysregulation of the balance between proliferation and apoptosis of endometrial cells in PCOS, attributed to elevated levels of androgens, estrogens, and hyperinsulinemia, results in heightened susceptibility to alterations during cellular proliferation. Thus GnRH-a contributes to the success of FET by decreasing estrogen and androgen levels and preventing high LH levels to improve endometrial receptivity. The majority of studies have supported the utilization GnRH-a pretreatment for endometrial preparation in woman with PCOS, although some studies have argued to the contrary (*Liu et al., 2021*; *Luo et al., 2021*).However, these dissenting studies lack a strong and rational explanation. There is still a lack of definitive conclusion about the effect of down-regulation protocol in FET cycles of women with repeated implantation failure (RIF) (*Davar, Dashti & Omidi, 2020*; *Yang et al., 2016*). However, tubal factors account for the majority of female infertility, and depending on the research population, female infertility caused by tubal factors might be as high as 67%. Additionally, a considerable portion of infertile couples are caused by male factors including oligospermia and asthenospermia (*Carson & Kallen, 2021*). It remains unclear whether HRT + GnRH-a improves pregnancy outcomes in these populations.

The success rate of IVF usually decreases with maternal age, especially in the woman after the age of 35 and 40 (*McCoy, Nakajima & Bohler Jr, 2009*). Few studies have explored the efficacy of HRT-FET with GnRH-a down-regulation to prepare endometrium in women of different ages, especially those of advanced age. In this research, reproductive outcomes in HRT and GnRHa-HRT will be compared in the general female population of different ages.

## MATERIALS & METHODS

### Study design and the subjects

The study was approved by the Ethics Committee of The Second Affiliated Hospital and Yuying Children's Hospital of Wenzhou Medical University (Approval number: 2023-K-205-01). We retrospectively collected all frozen blastocyst transfer cycles carried out between January 2020 and July 2022 at the reproductive center of The Second Affiliated Hospital of Wenzhou Medical University. Inclusion criteria: endometrial preparation protocol using HRT cycle with or without GnRH-a pretreatment. Exclusion criteria were: adenomyosis or endometriosis; PCOS; RIF; severe intrauterine adhesions, endometrial thickness was less than 6 mm; natural cycle IVF/ICSI; characteristic information incomplete; reproductive outcomes lost. Based on the criteria listed above, 4,091 cycles were included in this study. Cycles were grouped by maternal age: group A (<35 years old) and group B (≥35 years old). Each group was further subdivided into two sub-groups based on the endometrial preparation methods: the HRT group and the GnRHa-HRT group.

### Endometrial preparation protocols

HRT group: Supplementation with estrogens was administered on day 2–5 of the menstrual cycle after the sex hormone test and ultrasonography showed no abnormalities. Endometrial thickness and morphology were regularly evaluated with vaginal ultrasonography. Monitor the levels of E2, progesterone, and LH, and adjust the drug usage and dosage when necessary. The timing of endometrial transformation was determined by the clinicians. If endometrial thickness and morphology were satisfactory and serum progesterone level <1.5 ng/ml, progesterone was added to achieve endometrial transformation. Blastocysts were transferred on the fifth day after transformation.

GnRHa-HRT group: 3.75mg long-acting GnRH agonist was injected on the 2nd–5th day of menstruation, and 28–30 days later, patients were examined for serum FSH, LH, and E2 levels, and an ultrasound was performed to evaluate endometrium. When pituitary downregulation reached standards: LH < 5U/L, FSH < 5U/L, E2 < 50 pg/ml, endometrial thickness < 5 mm, and the largest follicle diameter was ≤5–10 mm, and then the protocol was the same as for HRT.

### Definition of clinical outcomes

Serum $\beta$-hCG levels were measured on day 10–14 after embryo transfer and transvaginal ultrasound was performed 2 weeks after the $\beta$-hCG test to diagnose pregnancy. Serum $\beta$-HCG >5 mIU/mL indicated HCG positive. Pregnancy outcomes included: biochemical pregnancy (a positive $\beta$-hCG level without visualization of an intrauterine gestational sac),

clinical pregnancy (the presence of a gestational sac), miscarriage (a loss of pregnancy before 28 weeks of gestation), and singleton live birth (a live born infant after 28 weeks of gestation).

## Observation indicators
Primary observation indicator: live birth rate. Secondary observation indicators: clinical pregnancy rate and miscarriage rate.

## Statistical analysis
Data were analyzed using Graphpad Prism (Graphpad Prism 9 software). Data with a skewed distribution are described as median (interquartile range (IQR)). Mann–Whitney test was performed for group comparison. The chi-square test or Fisher exact test was used for categorical variables. Multivariate logistic regression analysis was performed to explore the factors associated with live birth. The covariates included endometrial preparation protocol (with or without GnRH-a), maternal age, maternal body mass index (BMI), history of intrauterine adhesions surgery, and number of embryos transferred. $P < 0.05$ was considered statistically significant.

## RESULTS
According to the inclusion criteria and exclusion criteria, a total of 4,091 cycles were enrolled in this retrospective study. There were 2,605 cycles in group A and 1,486 cycles in group B. In group A, 1,965 cycles underwent HRT and 640 cycles underwent a long-term GnRH-a plus HRT; In group B, 1,054 cycles received HRT and 432 cycles received GnRH-a + HRT (Fig. 1).

The baseline characteristics were described in Table 1. For women <35 years old, the median age for the HRT group and GnRH-a group were 30.6 and 31.3 years, respectively. Among women aged ≥35, the median age of the HRT group was 38.1 years and that of the GnRHa-HRT group was 38.3 years. Male age, female BMI, duration of infertility, infertility type, and fertilization type were comparable between the groups.

The double embryo transfer rate was higher in the GnRHa-HRT group in younger women (70.00% *vs* 65.55%, $P = 0.0381$). The prevalence of prior uterine adhesions was higher in the down-regulation group among women ≥35 years (19.44% *vs* 12.14%, $P = 0.0004$). Additionally, the protocol in the fresh cycle was statistically different between groups ($P = 0.0014$ and $P = 0.0040$, respectively).

Table 2 shows the clinical outcomes of FET. Women aged <35 years had higher clinical pregnancy and live birth rates than women aged ≥35 years. Among young women, the HRT group had higher clinical pregnancy and live birth rates and lower biochemical pregnancy and miscarriage rates than the GnRHa-HRT group, although none of the differences were statistically significant. In women of advanced age, there was no statistically significant difference in biochemical pregnancy and miscarriage rates between the GnRHa-HRT group and HRT group, but there was a substantial increase in clinical pregnancy (44.68% *vs* 36.91%, $P = 0.0059$) and live birth rates (31.94% *vs* 24.38%, $P = 0.0036$) following down-regulation.

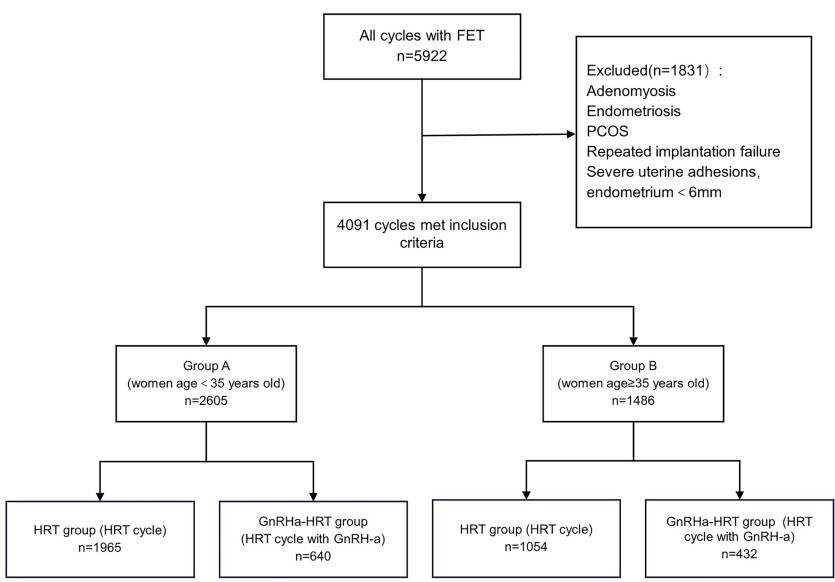

**Figure 1** **Selection and grouping of the study population.** FET, frozen embryo transfer; PCOS, polycystic ovary syndrome; HRT, hormone replacement treatment; GnRH-a, Gonadotropin-releasing hormone agonist.

Logistic regression analysis was performed to evaluate the variables associated with live birth rate (Table 3). Results showed that female age and number of embryos transferred had significant effects on the live birth rate in the whole population. The impact of downregulating or not on the live birth rate did not differ statistically among women aged <35 years (OR = 0.912, 95% CI [0.760–1.094], $P = 0.321$). However, down-regulation with GnRH-a prior to HRT significantly improved live birth rates in women of advanced age (OR = 1.614, 95% CI [1.246–2.091], $P < 0.001$).

## DISCUSSION

The main parameters influencing the clinical outcomes of FET are embryo quality, endometrial receptivity, and embryo-endometrial synchrony, and the most controllable of these is the endometrium. The optimal endometrial preparation regimen has been a topic of great concern. Currently, the most commonly used protocol is HRT. HRT cycles in the absence of GnRH-a suppression can result in a rise in LH. Pretreatment with GnRH-a prior to HRT can avoid the risk of spontaneous ovulation and terminates the implantation window in advance to increase endometrial receptivity (*El-Toukhy et al., 2004*). Whether GnRH-a should be used for down-regulation during FET cycles is not yet clear. Some studies considered down-regulation with GnRH-a has no significant effect on reproductive outcomes (*Davar, Dashti & Omidi, 2020*; *Liu et al., 2021*; *Luo et al., 2021*; *Prato et al., 2002*; *Samsami, Chitsazi & Namazi, 2018*; *Van de Vijver et al., 2014*). However, some studies argued to the contrary (*El-Toukhy et al., 2004*; *Hill, Miller & Frattarelli, 2010*; *Kang et al., 2018*; *Niu et al., 2013*; *Orvieto et al., 2008*; *Park et al., 2016*; *Wang et al., 2022*; *Yang et al., 2016*). However, most studies chose all FET cycles or special populations,

**Table 1  Demographic characteristics of patients between different age groups and endometrial preparation groups.**

| Characteristics | Group A (n = 2,605) | | | Group B (n = 1,486) | | |
|---|---|---|---|---|---|---|
| | HRT (n = 1965) | GnRHa-HRT (n = 640) | P value | HRT (n = 1054) | GnRHa-HRT (n = 432) | Pvalue |
| Maternal age (years) | 30.60 (28.20,32.60) | 31.30 (28.50,33.00) | 0.0016[*] | 38.10 (36.40,40.50) | 38.30 (36.60,40.68) | 0.2700 |
| Male age (years) | 32.00 (30.00,35.00) | 33.00 (31.00,35.00) | 0.1824 | 39.00 (37.00,42.00) | 39.00 (37.00,42.00) | 0.7680 |
| BMI (kg/m$^2$ ) | 21.09 (19.31,23.44) | 21.26 (19.56,23.81) | 0.0697 | 21.81 (20.06,23.92) | 22.03 (20.09,24.02) | 0.5264 |
| Duration of infertility (years) | 3.00 (1.00,4.00) | 3.00 (1.00,4.00) | 0.6756 | 3.00 (1.00,5.00) | 3.00 (1.00,5.00) | 0.6349 |
| Number of embryos transferred, n (%) | | | 0.0381[*] | | | 0.5588 |
| 1 | 677 (34.45%) | 192 (30.00%) | | 409 (38.80%) | 175 (40.51%) | |
| 2 | 1288 (65.55%) | 448 (70.00%) | | 645 (61.20%) | 257 (59.49%) | |
| Infertility type, n (%) | | | 0.4007 | | | 0.7348 |
| Primary | 754 (38.37%) | 258 (40.31%) | | 135 (12.81%) | 58 (13.43%) | |
| Secondary | 1211 (61.63%) | 382 (59.69%) | | 919 (87.19%) | 374 (86.57%) | |
| History of intrauterine adhesions, n (%) | 149 (7.58%) | 61 (9.53%) | 0.1320 | 128 (12.14%) | 84 (19.44%) | 0.0004[**] |
| Protocol in fresh cycle, n (%) | | | 0.0014[*] | | | 0.0040[*] |
| Agonist | 1566 (79.69%) | 488 (76.25%) | | 631 (59.87%) | 218 (50.46%) | |
| Antagonist | 23 (1.17%) | 0 | | 10 (0.95%) | 2 (0.46%) | |
| PPOS/LPOI | 212 (10.79%) | 96 (15.00%) | | 235 (22.30%) | 127 (29.40%) | |
| Others | 164 (8.35%) | 56 (8.75%) | | 178 (16.88%) | 85 (19.68%) | |
| Fertilization type, n (%) | | | 0.7898 | | | 0.3129 |
| IVF | 1515 (77.10%) | 485 (75.78%) | | 775 (73.53%) | 316 (73.15%) | |
| ICSI | 444 (22.60%) | 153 (23.91%) | | 265 (25.14%) | 114 (26.39%) | |
| IVF+ICSI | 6 (0.3%) | 2 (0.31%) | | 14 (1.33%) | 2 (0.46%) | |

**Notes.**

Abbreviations: PPOS, Progestin-primed ovarian stimulation; LPOI, luteal phase ovulation induction.

[*]$P < 0.05$.

[**]$P < 0.001$.

including adenomyosis or endometriosis, PCOS, severe uterine adhesions, and RIF. Some studies found that GnRH-a + HRT can improve the success rate of FET in these special people. Therefore, all patients with the above-mentioned disorders have been excluded from the study to avoid the influence of abnormal endometrial proliferation and abnormal
**Table 2 Comparison of reproductive outcomes.**

| Outcome | Group A (n = 2605) | | | Group B (n = 1486) | | |
|---|---|---|---|---|---|---|
| | HRT (n = 1,965) | GnRHa-HRT (n = 640) | P value | HRT (n = 1,054) | GnRHa-HRT (n = 432) | P value |
| Biochemical pregnancy rate | 282 (14.35%) | 107 (16.72%) | 0.1599 | 147 (13.95%) | 64 (14.81%) | 0.6826 |
| Clinical pregnancy rate | 1066 (54.25%) | 340 (53.13%) | 0.6480 | 389 (36.91%) | 193 (44.68%) | 0.0059[*] |
| Miscarriage rate | 182 (17.07%) | 69 (20.29%) | 0.1931 | 132 (33.93%) | 55 (28.50%) | 0.2203 |
| Live birth rate | 884 (44.99%) | 271 (42.34%) | 0.2522 | 257 (24.38%) | 138 (31.94%) | 0.0036[*] |

Notes.
[*]$P < 0.05$.

**Table 3 Logistic regression analysis of potential factors associated with live birth rate after FET.**

| Variable | Group A | | Group B | |
|---|---|---|---|---|
| | OR (95% CI) | P value | OR (95% CI) | P value |
| Endometrial preparation | | | | |
| HRT | Reference | | Reference | |
| GnRHa-HRT | 0.912 (0.760, 1.094) | 0.321 | 1.614 (1.246, 2.091) | <0.001[*] |
| Maternal age | 0.941 (0.916, 0.967) | <0.001[*] | 0.796 (0.757, 0.838) | <0.001[*] |
| BMI | | | | |
| $18.5 \leq BMI < 24.0$ | Reference | | Reference | |
| BMI < 18.5 | 0.973 (0.775, 1.222) | 0.814 | 0.664 (0.416, 1.060) | 0.086 |
| BMI ≥ 24 | 0.991 (0.814, 1.208) | 0.932 | 0.982 (0.739,1.306) | 0.902 |
| Duration of infertility | 0.986 (0.951, 1.023) | 0.463 | 0.985 (0.949, 1.023) | 0.443 |
| History of intrauterine adhesions | 0.769 (0.573, 1.033) | 0.081 | 0.737 (0.513, 1.060) | 0.100 |
| Number of embryos transferred | | | | |
| 1 | Reference | | Reference | |
| 2 | 1.315 (1.113, 1.555) | 0.001[*] | 1.505 (1.170, 1.936) | 0.001[*] |

Notes.
Abbreviations: OR, odds ratio; CI 95%, confidence interval.
[*]$P \leq 0.001$.

endometrial receptivity. Age is an independent factor affecting the pregnancy outcome of women. With the increase of age, female fertility decreases. With the increasing of maternal age, the quantity and quality of oocytes, embryo quality and pregnancy rate of FET exhibit a declining trend, and the incidence of embryo aneuploidy increases. Thus, endometrial preparation should be individualized for women of different ages.

In this study, we found that pituitary downregulation with GnRH-a before HRT cannot improve pregnancy outcomes in young women, which is consistent with previous studies. When the reason for infertility in young women was tubal, idiopathic, or male factors, the success rate of FET was similar between the two protocols (*Prato et al., 2002*). Also, in women aged less than 38 years, the implantation rate, clinical pregnancy rate, and live birth rate were comparable between the HRT group and GnRHa-HRT group after excluding specific factors such as adenomyosis, endometriosis and PCOS (*Xu et al., 2021*). We hypothesized that infertility caused by male or tubal factors would be more prevalent

in younger women, and hence GnRH-a down-regulation would have no significant effect on FET outcomes in these people.

Furthermore, the important finding of this research was that in women of advanced age, a significant improvement in pregnancy outcomes, including pregnancy and live birth rates, was identified in the GnRHa + HRT group. It contradicted the findings of a previous research, which indicated that GnRH-a combined with HRT didn't improve the reproductive outcomes of frozen-thawed embryo cycles in elderly patients, but led to a higher abortion rate (*Dong et al., 2020*). However, the study did not exclude the influence of abnormal endometrial proliferation in patients with PCOS, nor did it rule out patients with adenomyosis or endometriosis. These comorbidities can affect endometrial receptivity and pregnancy outcomes. In addition, a retrospective study found that, as compared to women aged 40–45 years, pretreatment with a long-acting GnRH-a improved the clinical and live birth rates in women <40 years (*Xu et al., 2023*). Different age selection and GnRH-a use timing could be the cause of the discrepancy in these results. The start of HRT in that study was after 14 days of GnRH-a, whereas the time of GnRH-a down-regulation in the present study was at least 28 days. Moreover, patients with PCOS and RIF were also not excluded from that study.

Analyzing the baseline characteristics of the patients we found that male age, female BMI and type of infertility were comparable between the groups. Advanced male age is associated with decreased semen quality such as decreased semen volume, decreased motile spermatids, and increased dysmorphic morphology, poorer DNA integrity, and chromosomal anomalies (*Bertoncelli Tanaka, Agarwal & Esteves, 2019*; *Kobayashi et al., 2017*; *Sharma et al., 2015*). All of above is related to ART failure and miscarriage (*Zini et al., 2008*). And GnRH-a can improve pregnancy outcomes in patients with male-factor infertility in FET cycles. It may be that it can facilitate dialogue between embryo and endometrium (*Yu et al., 2022*). It has been reported that pre-pregnancy BMI affects ART pregnancy outcomes. Among women undergoing FET, implantation rates, clinical pregnancy rates, and ongoing pregnancy rates were reduced in underweight women (*Kawwass et al., 2016*; *Tang et al., 2021*). This may be related to low leptin levels in underweight women, which regulate uterine angiogenesis and implantation (*Cervero et al., 2004*; *Veleva et al., 2008*). In addition, high BMI have an adverse effect on early pregnancy loss rate, live birth rate, clinical pregnancy rate and cumulative live birth rate (*Bakkensen, Strom & Boots, 2024*; *Cheng et al., 2024*; *Yang et al., 2021*; *Zheng et al., 2024*). The impairment of endometrial receptivity in obese women is related to displacement of the window of implantation and endocrine and metabolic disturbances induced by increased BMI (*Bellver et al., 2021*; *Pantasri & Norman, 2014*). A study has shown that partial obese patients may benefit from GnRH-a long protocol in fresh IVF-ET cycles, and one of the mechanisms may be that GnRH-a improves endometrial receptivity (*Wan et al., 2023*).

Also, previous studies demonstrated that patients with secondary infertility had a higher pregnancy probability compared with the patients with primary infertility (*Bushaqer et al., 2020*; *Stolwijk, Wetzels & Braat, 2000*; *Templeton, Morris & Parslow, 1996*). In our study, there was no statistically significant difference in male age, female BMI and type of infertility

between groups. Therefore, it was possible to avoid the interference of above-mentioned factors on results.

Importantly, we found previous uterus surgeries were more common in older women than in younger women, including induced abortion, mild adhesion separation surgery, and endometrial polyp removal. Inflammatory infections or a history of multiple uterine operations can cause uterine adhesions, leading to reduced uterine volume, endometrial damage, and lower endometrial blood perfusion, all of which can contribute to endometritis and alteration in the endometrial microenvironment (*Evans-Hoeker & Young, 2014*; *Mishra et al., 2016*). Inadequate perfusion of uterine blood flow and increased impedance to uterine artery and endometrial blood flow result in endometrial and subendometrial hypoxia, leading to reduced endometrial receptivity (*Wang et al., 2020*). Uterine artery impedance has a significant correlation with biochemical markers of uterine receptivity and can accurately predict the probability of pregnancy in FET cycles, as evidenced by a study indicating lower uterine artery pulsatility index in individuals achieving pregnancy compared to those who did not conceive in FET cycles (*Steer et al., 1995*). In our research, it was observed that in women aged 35 years and older, the rates of clinical pregnancy and live birth were significantly higher in the GnRHa-HRT group, despite a greater prevalence of previous intrauterine adhesions, which strengthened our conclusions.

A study confirmed that the endometrial flow index was higher in young women (*Engels et al., 2011*). Further, major changes in endometrial function occur after 35 years of age, and some of these mechanisms include cell cycle arrest, DNA repair inhibition, sugar metabolism, and immune responses (*Devesa-Peiro et al., 2022*). Importantly, GnRH-a can decrease the release of inflammatory factors, influence cytokine expression, and increase endometrial blood flow and energy metabolism to improve endometrial receptivity (*Chen et al., 2019*). Therefore, GnRH-a suppression may improve pregnancy outcomes in patients of advanced age through the above pathways.

In HRT protocol, exogenous estrogen and progesterone inhibit the growth of dominant follicles and regulate the growth and transformation of the endometrium. But there is still a chance of premature rise of LH to disturb embryo implantation. Pituitary suppression with GnRH-a not only avoids dominant follicle development but also reduces the adverse effects of premature rise of LH on the "implantation window" and embryo implantation (*El-Toukhy et al., 2004*).

Premature rise of LH is more common in old female patients (*Martinez et al., 2002*). By inhibiting the hypothalamus-pituitary-ovary (HPO) axis, GnRH-a can reduce LH levels to improve endometrial receptivity.

Furthermore, as the pelvic environment declines in older women, inhibiting pelvic inflammatory factor production by pretreatment with GnRH-a may be beneficial for pregnancy. Despite the decreased embryo quality in older women, pregnancy outcomes were significantly improved after down-regulation, presumably because GnRH-a encouraged embryo-endometrial interactions during early embryo implantation (*Raga et al., 1998*).

Multivariate logistic regression analysis showed that the success of FET cycles was significantly influenced by the number of embryos transferred in addition to the women's

age. The studies have shown that double embryo transfer (DET) increased the chance of clinical pregnancy and live birth compared with single embryo transfer (SET), while also increasing the risk of multiple pregnancies and preterm births (*Kamath et al., 2020*; *Long et al., 2020*; *Roberts et al., 2011*). For young women, SET should be incorporated into clinical practice owing to the reduction in pregnancy and neonatal complications (*Ma et al., 2022*; *Wu et al., 2022*). In women of advanced age, the recommended number of embryos to be transferred is controversial (*Ma et al., 2022*; *Tannus et al., 2017*). Clinically, the number of embryos transferred is usually determined according to the patient's age, uterine surgery history, the number of available embryos, and the patient's personal desires. In our study, there was no difference in the number of embryos transferred in women ≥35 years old. The double embryo transfer rate was higher in the GnRHa-HRT group in younger women, but the clinical pregnancy rate and live birth rate were comparable to those in HRT group, which strengthened the interpretation of our findings that GnRH-a improves clinical pregnancy and live birth rates in older women undergoing HRT but not in younger women. Also, multivariate logistic analysis showed that previous intrauterine adhesions had no statistically significant effect on the live birth rate, the reason may be that in our study, cycles with endometrial thickness less than six mm have been excluded to prevent interference of endometrial thickness on the results.

To avoid the effect of abnormal endometrial proliferation and receptivity, the present study excluded patients with adenomyosis, endometriosis, PCOS, severe uterine adhesions, and RIF. The effect of GnRH-a down-regulation prior to HRT in women of all ages, however, has received very little research in the general population. The magnitude of our sample size boosted the accuracy of our findings. There were some limitations to the present. This article was a retrospective study and lacked validation from prospective studies. The retrospective study design may lead to selection bias. Although logistic analysis adjusted for confounder effects, it could not control for all factors. The retrospective data was collected from single center, large, prospective, multi-centered, randomized controlled trials under strict criteria are required.

In conclusion, clinicians should take the women's age into account when preparing the endometrium during FET cycles. A general hormone replacement treatment protocol is available for young female patients without adenomyosis, endometriosis, PCOS, severe uterine adhesions, and RIF to reduce the frequency of visits and expenditures. In women of advanced age, it is even more difficult to obtain embryos, thus GnRH-a down-regulation along with HRT may be tried to improve the rate of pregnancy.

## ACKNOWLEDGEMENTS

The authors would like to thank the staffs in the reproductive center of the Second Affiliated Hospital and Yuying Children's Hospital of Wenzhou Medical University for their help.

### Funding

The authors received no funding for this work.

## Competing Interests

The authors declare there are no competing interests.

## Author Contributions

- Jianghuan Xie conceived and designed the experiments, performed the experiments, analyzed the data, prepared figures and/or tables, authored or reviewed drafts of the article, and approved the final draft.
- Jieqiang Lu conceived and designed the experiments, performed the experiments, authored or reviewed drafts of the article, and approved the final draft.
- Huina Zhang conceived and designed the experiments, performed the experiments, authored or reviewed drafts of the article, and approved the final draft.

## Human Ethics

The following information was supplied relating to ethical approvals (*i.e.*, approving body and any reference numbers):

The study was approved by the Ethics Committee of The Second Affiliated Hospital and Yuying Children's Hospital of Wenzhou Medical University (Approval number: 2023-K-205-01).

## Data Availability

The raw measurements are available in the Supplementary File.

## Supplemental Information

Supplemental information for this article can be found online at http://dx.doi.org/10.7717/peerj.17447#supplemental-information.

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
