# Peer review of "Effect of GnRH agonist down-regulation combined with hormone replacement treatment on reproductive outcomes of frozen blastocyst transfer cycles in women of different ages"

_PeerJ, doi:10.7717/peerj.17447_

## Round 0.1 · original submission · Major Revisions

Please address the concerns of both reviewers and amend the manuscript accordingly.

Reviewer 1 ·

Basic reporting

no comment

Experimental design

1. The exclusion criteria in the article include patients with PCOS. Please briefly explain the basis for exclusion. The combination of downregulation and hormone replacement cycle has a clear benefit for the population with EMT, while there are more confounding factors for endometrial preparation failure in the RIF population. These two groups of people do indeed interfere with the results of data analysis and should be excluded. But PCOS itself seems to have little impact on the outcome of endometrial preparation, and there is still controversy in the introduction section about the benefits of this plan in pregnancy outcomes in the PCOS population. Is there a need for exclusion?
2. Abstract section: The Result indicates that the increase in live birth rate of the downregulated hormone replacement regimen only occurs in the elderly female population, while the Conclusion expresses it as "Among the general population", which is not very reasonable
3.The " Women aged <35 years had lower clinical pregnancy and live birth rates than women aged >35 years " in Result appears to be the opposite result in Table 2.
4. In the discussion, the author mentioned that the elderly population has a more common history of uterine adhesions compared to the younger population, and GnRHa may improve the endometrial environment by reducing the release of inflammatory factors, increasing endometrial blood flow, and increasing pregnancy rates. However, in the elderly population, the GnRHa HRT group itself has a lower history of uterine adhesions than the HRT group, so the statistical differences in pregnancy outcomes seem to require further discussion of the influence of confounding factors.

Validity of the findings

no comment

Additional comments

no comment

Reviewer 2 ·

Basic reporting

This study provides a comprehensive overview of the enrolled cycles and baseline characteristics, there is a lack of discussion regarding potential biases introduced by the retrospective design.

Experimental design

Retrospective studies inherently carry risks of selection bias and incomplete data capture, which could impact the robustness of the findings.

Validity of the findings

In Table 1, the description of baseline characteristics is informative, but there is limited discussion on how these characteristics might influence the outcomes of interest. Further analysis or discussion on how variables such as male age, female BMI, and infertility type might confound or interact with the treatment protocols could enhance the interpretation of the results.

Although the study reports differences in double embryo transfer rates and prevalence of prior uterine adhesions between groups, the clinical significance of these findings is not thoroughly explored. Providing context on the potential impact of these differences on overall treatment success or patient outcomes would strengthen the interpretation.

Please address the formatting typo of P-value of group A in Table 2.

In Table 3, the logistic regression analysis highlights significant variables associated with live birth rates, yet the interpretation of these findings lacks depth. Further discussion on the clinical implications of the identified associations would enhance the discussion of these results.

The Results and Discussion sections would benefit from a more critical examination of the limitations inherent in the study design and methodology. Addressing limitations such as retrospective data collection, potential confounding variables, and the generalizability of findings could improve the overall interpretation of the study results.

---

## Round 0.2 · accepted · Accept

All the issues pointed by the reviewers were addressed and the revised version is acceptable now.

Reviewer 2 ·

Basic reporting

Thanks for the author's responses and the revised manuscript looks good to me.

Experimental design

no comment

Validity of the findings

no comment

Additional comments

no comment